# A Reflection and Outlook on Clinical Adaption of Large Language Models

**Hanyin Wang,**[1] **Chufan Gao,**[2] **Jimeng Sun**[2,3]

[1] Division of Hospital Internal Medicine, Mayo Clinic Health System, Mankato, Minnesota
[2] Department of Computer Science, University of Illinois Urbana-Champaign, Champaign, Illinois
[3] Carle Illinois College of Medicine, University of Illinois Urbana-Champaign
wang.hanyin@mayo.edu

## Introduction

The recent advancements in large language models (LLMs) have brought about a significant revolution in various aspects of natural language processing (NLP). The emergence of potent open-source LLMs has paved the way for domain-specific fine-tuning within the clinical field. A recent survey comprehensively summarized the latest applications in constructing clinical LLMs, highlighting both their challenges and applications (Zhou et al. 2023b). In this study, we aim to build upon this previous work and provide further in-depth analysis into existing clinical LLMs, with a focus on their domain adaption approaches. Our objective is to stimulate meaningful discussions among participants during the AAAI workshop. We believe that by delving into these aspects, we can contribute to a better understanding of the potential and limitations of clinical LLMs.

## Methods

We compared a selection of recently published clinical LLMs that focused on medical knowledge injection in Table 1. These works encompassed either domain-specific tuning of open-source LLMs or pretraining from scratch using medical corpora. We have excluded models lacking published documentation detailing their training processes.

Notably, varying factors — model size, prompting strategy, and the use of proprietary versus open-source models — significantly complicate the process of directly comparing performance across different models. Despite these challenges, our analysis yielded several interesting observations.

## Results

**Top performers in MedQA and PubMedQA** The MedQA test is frequently used as a benchmark for medical knowledge. The leading model of Palmyra-Med was developed by simply fine-tuning the proprietary Palmyra-40B model on the training sets of PubMedQA and MedQA . This approach resulted in a significant improvement from the baseline (43.1 to 72.4 in MedQA, and 64.1 to 81.1 in PubMedQA). However, the reason for its improvement is difficult to discern, given its non-open-source nature and the employment of standard task-specific tuning steps.

PubMedQA serves as a benchmark for testing reasoning abilities over biomedical research texts. The leading model on this task, Meditron-70B, underwent the largest amount of continuous pretraining on the PubMed corpus. Notably, a much smaller model, BioGPT, pretrained with PubMed abstracts from scratch achieved comparably high performance.

**Continuous pretraining vs. supervised fine-tuning (SFT)** Prior research suggests that the majority of LLM's knowledge is acquired during the pretraining phase (Zhou et al. 2023a). Consistent with this, we observed robust performances from models that underwent continuous pretraining on domain-specific corpora. Notably, several studies employed instruction tuning based on question-answering pairs or dialogues infused with medical knowledge (e.g., curated from medical knowledge graphs). This specialized approach to SFT has demonstrated effectiveness, as evidenced in models like Clinical Camel and PMC-LLaMA. An intriguing finding can be seen from the ablation studies of PMC-LLaMA, which engaged in both continuous pretraining and SFT. The most substantial improvement in MedQA scores occurred during the instruction tuning phase. However, this improvement was only observed after the models had undergone the pretraining phase first.

**Model size vs. compute power** In practical scenarios, developers are often constrained by fixed computing budgets. Assuming that achieving the highest benchmark score is the ultimate goal (which may not be realistic in real-world applications), the efficiency of different approaches can vary significantly. For instance, the gain in PubMedQA performance per compute unit for BioMedGPT-LM is almost 40 times higher than that of Meditron (See Appendix). This difference is largely attributable to the smaller model size of BioMedGPT-LM.

**Training data** The majority of the models employed the PubMed corpus, supplemented with additional sources such as medical textbooks, clinical guidelines, and medical knowledge graphs. However, there appears to be a lack of a rigorous selection process for high-quality medical data. Notably, the sole LLM trained from scratch using electronic medical records (EMR) data from real patients, GatorTronGPT, scored the lowest in both MedQA and PubMedQA. This finding suggests that EMR data alone may not possess adequate medical knowledge.

**Table 1: Overview of clinical LLMs on MedQA and PubMedQA performance**

| Model | MedQA | PudMedQA | Training data | Approach |
|---|---|---|---|---|
| Palmyra-Med (Kamble and AlShikh 2023) | 72.4 | 81.1 | 160K data from traing set of PubMedQA and MedQA | SFT on Palmyra 40B |
| Meditron (Chen et al. 2023) | 64.4 | 81.6 | 47B tokens of PubMed data plus clinical guidelines | CP (1 epoch) on LLaMA-2 70B |
| Clinical Camel (Toma et al. 2023) | 60.7 | 77.3 | 174K multi-step dialogues pairs for knowledge encoding | SFT on LLaMA-2 70B with Qlora (5 epochs) |
| PMC-LLaMA (Wu et al. 2023) | 56.4 | 77.9 | 8B tokens of medical literature for CP, 200M tokens of medical question answering pairs for SFT | CP (3 epochs) and SFT (5 epochs) on LLaMA 13B |
| BioMedGPT-LM (Luo et al. 2023) | 50.4 | 76.1 | 26B tokens from from PubMed data | CP (1 epoch) on LLaMA-2 7B Chat |
| GatorTronGPT (Peng et al. 2023) | 45.1 | 77.6 | 82 billion words from EMR and 195 billion words from general domain | Pretrain from scratch (1 epoch), GPT-3 architecture with 30B |
| BioGPT (Luo et al. 2022) | - | 81.0 | 15M PubMed abstracts | Pretrain from scratch (1 epoch), GPT-2 architecture with 1.5B |
| AntGLM-Med (Li et al. 2023) | - | 80.6 | 15B tokens of medical literature for CP, 632K medical instruction data for SFT | CP and SFT on AntGLM-10B |
| LLaMA-2-7B | 29.1 | 49.1 | | |
| LLaMA-2-70B | 49.0 | 72.8 | | |
| MedPaLM-2 | 85.4 | 81.8 | | |
| GPT-4 | 78.6 | 82.0 | | |

For each model, only listed best performance. Base model performance of LLaMA-2, MedPaLM-2 and GPT-4 obtained from (Chen et al. 2023) and respective task's leaderboard. CP: continued pretraining. SFT: supervised fine-tuning.

## Discussions

**Medical knowledge injection** The models discussed in this study primarily focus on medical knowledge injection, employing two main approaches: continued pretraining and SFT. A recent study suggested that retrieval-augmented generation (RAG) consistently outperforms continued pretraining for knowledge injection. However, that study did not incorporate downstream instructional tuning, which, as demonstrated in the case of PMC-LLaMA (Wu et al. 2023), may be an essential step in activating the knowledge acquired during pretraining. Future research could compare SFT with continued pretraining to evaluate computational efficiency and effectiveness. Additionally, integrating RAG with these methods could offer further insights.

**Training data selection** It remains unclear whether the extensive corpus of PubMed literature necessarily constitutes "high-quality" data for the purpose of medical knowledge injection. For instance, a well-written clinical guideline published a decade ago might contain outdated treatment recommendations. Incorporating such data into training could be inaccurate and misleading to the model. An intriguing question pertains to the nature of knowledge embedded in EMR: it may reflect the peculiarities of clinical documentation, which is known to exhibit significant variation in writing style among providers, or it might encompass medical knowledge. As evidenced by the GatorTronGPT study (Peng et al. 2023), one could hypothesize that injecting medical knowledge using clinical notes is less effective compared to training from dedicated medical knowledge sources.

**Copyright concern** A proportion of the resources included in the commonly utilized PubMed corpus may be subject to non-transferable licenses (IDSA 2024). In the wake of the recent lawsuit, The New York Times vs. OpenAI (NYT 2023), it becomes crucial to consider the implications of using such unauthorized materials in model training. A promising alternative approach could involve the utilization of synthetic data generated by domain-adapted LLMs for future applications.

**Downstream use cases** Although much of the current research focuses on medical knowledge benchmarks, it remains uncertain whether a clinical foundation model would excel in knowledge-intensive downstream tasks. In enterprise use cases, LLM might be more suitable for tasks that can accommodate a higher tolerance for error and inaccuracy. Potential examples within the clinical domain could include administrative procedures (e.g., insurance appeals), patient flow processes (e.g., scheduling appointments), or clinical documentations. Bridging the gap between possible clinical applications and academic research efforts is crucial.

## Conclusion

The early adoption of foundation models in the clinical domain is encouraging. However, many open questions remain to be addressed in this rapidly evolving field.

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

# Appendix

## Calculate benchmark improvements per compute unit

The basic equation giving the cost to train a transformer model is given by:

$$C = 6PD$$

where:

- $C$ is the compute required to train the transformer model, in total floating point operations
- $C = C_{\text{forward}} + C_{\text{backward}}$
- $C_{\text{forward}} \approx 2PD$
- $C_{\text{backward}} \approx 4PD$
- $P$ is the number of parameters in the transformer model
- $D$ is the dataset size, in tokens

Subsequently, we can calculate a simple Score Improvement per Compute Unit (SICU) using the improvement of benchmark score and the total compute.

$$\text{SICU} = \frac{\Delta \text{Benchmark Score}}{C}$$

Based on Table 1, PubMedQA SICU is 0.0027 for Meditron, and 0.1071 for BioMedGPT-LM.