# OpenReview forum: "A Reflection and Outlook on Clinical Adaption of Large Language Models"
_AAAI.org/2024/Spring_Symposium_Series/Clinical_FMs — AAAI 2024 SSS on Clinical FMs_

### Official Review · Reviewer_o7is · 2024-02-20
**Good review on clinical LLM reasoning benchmarks, and provide insight of training data selection and domain adaptation approaches.**

**Rating:** 7
**Confidence:** 4

**Review:**

This paper explores the use and adaptation of large-scale language modeling (LLM) to the clinical domain. It builds on previous work and provides an in-depth analysis of existing clinical LLMs, focusing on domain adaptation approaches. The study compares various models according to their medical knowledge infusion strategies, including domain-specific adaptation and pre-training from scratch using a medical corpus. The study emphasizes the effectiveness of continuous pre-training and supervised fine-tuning in improving model performance on MedQA and PubMedQA medical benchmarks. The discussion raises important points about the selection of training data, the potential for retrieval enhancement generation, and the need for careful consideration of copyright issues. The discussion also touches on the utility of clinical LLM in downstream applications and the challenges that remain in bridging the gap between clinical needs and academic research.
While there are areas for further research and clarification, the study stimulates meaningful discussion on the potential and limitations of clinical LLMs, paving the way for future advancements.

---

### Official Review · Reviewer_EtCU · 2024-02-22
**Good study focusing a comprehensive analysis of current clinical Large Language Models (LLMs)**

**Rating:** 6
**Confidence:** 4

**Review:**

This paper provides a comprehensive analysis of the current state and future prospects of clinical Large Language Models (LLMs), focusing on domain adaptation strategies, performance comparisons, and the potential impact on healthcare. The following points should be considered:
1. How did the authors ensure the fairness of the results reported in Table 1, given the varying sizes, training data, and training methodologies employed by these models? Especially the differences in the evaluation settings on these two datasets, for example, BioGPT and GatorTronGPT employed prefix-tuning/prompt tuning to evaluate the pretrained LLMs, where the parameters of the LLMs are kept frozen, while PMC-LLaMA and BioMedGPT employed instruction tuning/fine tuning on the target dataset, where the parameters of the LLMs are trainable and may be better adapted to the downstream tasks.

2. The results in Table 1 are informative but could be enhanced, for example, the performance metric should be clarified, the column name "Approach" should be "Training approach", and the unit of parameter size should be clarified (e.g., 70B ->70 Billion). In the " Training data" column, there is some confusion about the unit (e.g., 160K data, 47B tokens). It would be beneficial to add a column to report the computational cost for each model.

3. Could the authors elaborate on the ethical and legal implications of using potentially copyrighted or sensitive patient data in training clinical LLMs?

---

### Official Review · Reviewer_cr9S · 2024-02-23
**Reject**

**Rating:** 4
**Confidence:** 4

**Review:**

The paper attempts to present an in-depth analysis of the current state and future directions of clinical Large Language Models (LLMs). The authors compare recent clinical LLMs, focusing on medical knowledge injection and domain-specific tuning or pretraining approaches. They provide insights into model performance across various benchmarks, including MedQA and PubMedQA, and discuss the implications of continuous pretraining versus supervised fine-tuning, model size, compute power, and the quality of training data.

**Pros**
1) **Easy to follow**: The paper is easy to follow but has some issues (pointed out in Cons section)
2) **Authors Cover various aspects of Clinical LLMs**:

    2a. The training data and approach are well summarised in Table 1.

    2b. The Results section provides insightful comments on continuous pretraining versus supervised fine-tuning, model size versus compute power, and the quality of training data.

**Cons**

1) **Evaluation in Table 1**: It's unclear whether the authors conduct the evaluations themselves or if the numbers are reported from respective papers. The table caption states, "For each model, only listed best performance," but it's not clear what "best performance" refers to. Additionally, the metrics used for performance (e.g., accuracy) should be clearly mentioned in the table.

2) **Some excerpts need to be refined:**

     2a. The conclusion appears to be very brief.

     2b. The discussion on downstream use cases is not comprehensive. Important industrial downstream tasks, such as early prevention of diseases and personalized medicine based on patient history, are not adequately covered.

4) **Novelty:** The paper seems to add little value to existing surveys [1, 2, 3]. Many of the discussed points, such as fine-tuning versus pre-training, training data, results on MedQA and PubMedQA, along with applications of LLMs in downstream use cases, are already extensively covered in the cited survey papers.

5) **Lack of Related Works**: Significant related works [2, 3] are not mentioned, which is a notable omission.

[1] Zhou, H.; Gu, B.; Zou, X.; Li, Y.; Chen, S. S.; Zhou, P.; Liu, J.; Hua, Y.; Mao, C.; Wu, X.; et al. 2023b. A survey of large language models in medicine: Progress, application, and challenge. arXiv preprint arXiv:2311.05112

[2] He, Kai, et al. "A survey of large language models for healthcare: from data, technology, and applications to accountability and ethics." arXiv preprint arXiv:2310.05694 (2023).

[3] Singhal, Karan, et al. "Large language models encode clinical knowledge." Nature 620.7972 (2023): 172-180.

While the authors mention results and discussions insightfully, they lack novelty, and most points exist already in the survey papers pointed out above. Additionally, the missing evaluation details in Table 1 are a concern.